# Management of Intra-Abdominal Infections: The Role of Procalcitonin

**DOI:** 10.3390/antibiotics12091406

**Published:** 2023-09-04

**Authors:** Brian W. C. A. Tian, Vanni Agnoletti, Luca Ansaloni, Federico Coccolini, Francesca Bravi, Massimo Sartelli, Carlo Vallicelli, Fausto Catena

**Affiliations:** 1Department of General Surgery, Singapore General Hospital, Outram Road, Singapore 169608, Singapore; briananthonytian@gmail.com; 2Anesthesia and Intensive Care Unit, M. Bufalini Hospital, 47521 Cesena, Italy; 3Department of Surgery, Pavia University Hospital, 27100 Pavia, Italy; 4General, Emergency and Trauma Surgery Department, Pisa University Hospital, 56126 Pisa, Italy; 5Santa Maria Delle Croci Hospital, 48121 Ravenna, Italy; 6Department of Surgery, Macerata Hospital, Via Santa Lucia 2, 62100 Macerata, Italy; 7Department of Emergency and Trauma Surgery, M. Bufalini Hospital, 47521 Cesena, Italy; 8Department of Surgery, “Maurizio Bufalini” Hospital, 47521 Cesena, Italy

**Keywords:** procal, procalcitonin, sepsis, intra-abdominal, infection

## Abstract

Patients with intra-abdominal sepsis suffer from significant mortality and morbidity. The main pillars of treatment for intra-abdominal infections are (1) source control and (2) early delivery of antibiotics. Antibiotic therapy should be started as soon as possible. However, the duration of antibiotics remains a matter of debate. Prolonged antibiotic delivery can lead to increased microbial resistance and the development of nosocomial infections. There has been much research on biomarkers and their ability to aid the decision on when to stop antibiotics. Some of these biomarkers include interleukins, C-reactive protein (CRP) and procalcitonin (PCT). PCT’s value as a biomarker has been a focus area of research in recent years. Most studies use either a cut-off value of 0.50 ng/mL or an >80% reduction in PCT levels to determine when to stop antibiotics. This paper performs a literature review and provides a synthesized up-to-date global overview on the value of PCT in managing intra-abdominal infections.

## 1. Introduction

Patients with intra-abdominal sepsis suffer from significant mortality and morbidity. The main pillars of treatment for intra-abdominal infections are (1) source control and (2) early delivery of antibiotics [1,2]. In the 19th century, patients with intra-abdominal sepsis had an almost 90% mortality rate. In recent years, better and earlier surgical intervention and utilization of antibiotics have reduced that rate to about 30% [3,4,5,6,7,8].

It is well accepted that, upon clinical recognition of intraabdominal sepsis, the patient must receive antibiotics immediately. However, there is still much debate regarding the duration of antibiotics [8]. Prolonged antibiotic use will encourage greater amounts of microbial resistance and the development of nosocomial infections [8]. There has been much research on biomarkers and their ability to help decide the duration of antibiotics [9,10]. Examples of these biomarkers include procalcitonin (PCT), interleukins, C-reactive protein (CRP), etc.

Procalcitonin is synthesized by the C-cells of the thyroid. It is the prohormone of calcitonin. Its production is correlated to the presence of microbial endotoxins and indirectly responds to cytokines (e.g., IL-1, IL-6 and TNF) [11]. There has been an increase in the number of studies studying PCT’s role as a biomarker. Several studies have reported that when PCT is lower than 0.50 ng/mL or when its levels are reduced by at least 80% of the initial value [12,13], clinicians can stop the antibiotics with good outcomes [14,15,16,17].

In general, PCT reaches its highest levels 24–48 h after an infection. Proper administration of antibiotics and good source control can decrease PCT values by approximately 50% within 24–36 h. This trajectory has even been demonstrated in renal failure patients, as PCT accumulation does not occur [18]. Research has also shown that if the infectious stimulus is removed, the concentration of PCT decreases steadily and eventually returns to normal ranges on days 5–7 [19]. Overall, the evidence of PCT utility is growing and evolving constantly, and thus far, PCT appears to be very useful in determining the duration of antibiotic administration [9,20,21].

We aimed to perform a literature review and provide a synthesized up-to-date global overview on the utility and role of PCT in managing intra-abdominal sepsis.

## 2. Methods and Materials

Two authors (BT, FC) searched several databases: PubMed, MEDLINE, the Cochrane Library and EMBASE. Keywords for searching included biomarkers, procal, procalcitonin, inflammatory mediators, abdominal sepsis and intra-abdominal infection. Both authors (BT, FC) reviewed all articles. The search was restricted to the English language. Only literature reviews, meta-analyses, and original studies (cohort studies, trials) were included.

## 3. Discussion

### 3.1. Procalcitonin (PCT) versus C-Reactive Protein (CRP)

PCT and CRP are among several biomarkers of inflammation and sepsis. In normal physiological states, PCT’s serum concentrations are negligible [22]. The exact biomechanism that leads to the production of PCT, and PCT’s relationship with inflammation is not fully understood. At present, it is accepted that PCT is synthesized in mononuclear cells and the liver [23,24], and that lipopolysaccharides and cytokines serve as its stimulus for production. Conversely, CRP is produced by the liver, in response to interleukin-6 (IL-6). In turn, IL-6 is produced during infection and inflammation [25].

Simon et al. [26] analyzed 10 studies (905 patients) and showed that PCT was significantly better at stratifying between bacterial and non-infective causes of inflammation, as compared to CRP. There was a statistically significant difference between the pooled sensitivity analyses for PCT [88% (80–93% CI)] when compared against CRP [75% (62–84% CI)]. PCT [81% (67–90%CI)] also had a statistically significant higher pooled specificity when compared against CRP [67% (56–77%)]. Similar trends were observed when comparing sensitivity and specificity of PCT against CRP in identifying bacterial versus viral infections.

These data suggest that PCT is potentially a better biomarker than CRP. Nonetheless, the interpretation of PCT results must be performed in the context of appropriate clinical judgement. In these nascent days of PCT adoption and utilization, serial measurements of PCT can perhaps be used to exclude a diagnosis of systemic sepsis, rather than to rule it in. There is also increasing evidence for its use as an antibiotic stewardship tool to reduce the duration of antibiotic exposure [27].

### 3.2. Procalcitonin Is Useful in the Management of Complicated Appendicitis

It can be helpful to consider two broad categories of acute appendicitis:

(1) Uncomplicated acute appendicitis (UAA) and (2) complicated acute appendicitis (CAA). UAA refers to simple, suppurative appendicitis (without gangrene, perforation or abscess). Conversely, CAA refers to appendicitis with evidence of gangrene, perforation or abscesses [28].

In today’s clinical practice, the preferred treatment of choice for appendicitis remains an upfront appendicectomy. However, recent studies have shown that for patients with UAA, when comparing between conservative treatment with antibiotics and surgical treatment, there were no significant differences in terms of complications, length of stay or antibiotic duration [29,30,31,32]. Although treatment with antibiotics alone in acute appendicitis is still an unpopular choice, the potential for such conservative treatment has necessitated efforts to explore methods to differentiate between UAA and CAA upon admission.

Current imaging technologies such as ultrasound, CT, or MRI enable physicians to diagnose acute appendicitis and to assess severity. However, it is only easy to assess severity if the appendix has already perforated or an abscess has formed. Current imaging technologies have limitations in attempting to detect gangrenous changes.

Research has shown that certain biomarkers could be employed to supplement the clinical diagnosis of appendicitis. Some of these markers are CRP, urinary 5-HIAA, PCT, serum amyloid A, IL-6, D- white cell and neutrophil counts [33,34,35,36,37,38,39]. However, there still remains no single reliable biomarker that can definitively identify UAA from CAA cases [40,41].

Some studies have shown that PCT can be helpful in diagnosing acute appendicitis [33,42,43]. Abbas et al. [33] reported that, in adults, PCT levels on admission had an 85% sensitivity and 74% specificity for diagnosing acute appendicitis. Chandel et al. [42] reported that, in children, PCT had a 95.65% sensitivity and 100% specificity for diagnosing appendicitis. This was corroborated by the findings of Khan et al. [43]. Some studies have suggested that PCT levels in children could even stratify between UAA and CAA [44,45,46,47].

Conversely, there have been studies that suggested that PCT had no ability to diagnose appendicitis in adults or children [34,38,48]. Sand et al. [48] demonstrated that PCT only had a 14% sensitivity in diagnosing appendicitis, making it a poor diagnostic tool. However, Sand also reported that the values of PCT escalated in patients with perforated or gangrenous appendicitis. Wu et al. [49] also demonstrated that PCT was not useful for screening cases with suspected appendicitis. Wu et al. however did find that PCT levels were significantly raised in CAA patients. When a cut-off value of 0.5 ng/mL was used, PCT had a sensitivity of 29% and specificity of 95%. Wu et al. therefore suggested that perhaps PCT levels could be useful in detecting patients with suspected CAA.

Li et al. [50] demonstrated that PCT levels were significantly elevated in patients with CAA, versus those with UAA. In that paper, PCT, age and CRP were all positively correlated. A ROC analysis demonstrated an AUC of 0.987 (0.965–1.000) for PCT when diagnosing for CAA. A cut-off value of 0.42 ng/mL was applied and demonstrated a 100% sensitivity and 95.1% specificity. PCT levels were shown to be an independent factor for CAA, even after adjusting for age and CRP.

Tanrikulu et al. [51] had also shown that PCT and CRP levels were significantly elevated in patients with CAA. However, neither biomarker was able to accurately predict the pathological changes.

### 3.3. PCT May Be Able to Predict the Severity of Cholecystitis and the Likelihood of a Difficult Laparoscopic Surgery and Higher Risk for Post-Operative Complications

The Tokyo Guidelines [TG] 07, 13 and 18, provided basic criteria to help grade the severity of acute cholecystitis [52,53,54]. Mild and moderate cholecystitis patients can be candidates for laparoscopic cholecystectomy. Conversely, severe cases should be considered for upfront percutaneous cholecystostomy [54], as laparoscopic cholecystectomy is expected to be difficult due to edema and inflammatory adhesions [54]. Hence, the ability to predict the severity of cholecystitis would very helpful, as it impacts decision making.

Yuzbasioglu et al. had shown that PCT can help stratify cases of acute cholecystitis according to severity [55]. Yuzbasioglu described a positive correlation between PCT and CRP and leukocyte levels. Sakalar et al. also showed that PCT level was strongly correlated to acute cholecystitis severity [56].

However, identifying the clinical severity of acute cholecystitis is still different from predicting the difficulty of the laparoscopic cholecystectomy (LC), although one is thought to extrapolate to the other. In that regard, the TG criteria are still unable to effectively predict the difficulty of laparoscopic cholecystectomy [8]. A difficult laparoscopic case means that the surgeon will face prolonged operating times, increased risk of biliary tract injuries, or higher likelihood of conversion to open surgery [25].

There are several studies analyzing factors associated with the technical difficulty of LC [57,58,59,60]. However, no study to date has found a reliable and consistent predictor for a difficult LC. As discussed in TG18, one of the major considerations of a difficult case, is the severity and degree of inflammation in the Calot’s triangle [61]. Wu et al. suggested that, therefore, the degree of difficulty in LC might be correlated to the extent of intra-abdominal inflammation in these patients [62]. Wu studied the relation between PCT levels and the technical difficulty of an LC case. In Wu’s study, the AUC for the ROC curve of PCT was 0.927 [95% CI 0.882–0.973 (*p* < 0.001)]. Wu concluded that, when the preoperative PCT levels were >1.50 ng/mL, the surgeon was likely to experience a technically challenging case. Overall, Wu demonstrated that PCT had a 91.3% sensitivity and 76.8% specificity for predicting a difficult LC [62].

### 3.4. PCT Can Be Used to Manage Antibiotic Therapy in Acute Diverticulitis

Biomarkers such as CRP and calprotectin have thus far only demonstrated limited specificity and sensitivity in the management of diverticulitis [63]. Jeger et al. postulated that procalcitonin could play a role, because the management and clinical picture of diverticulitis bear similarities to respiratory conditions [64]. Most notably, most cases of diverticulitis are of viral origin, similar to respiratory infections. Since tracking PCT has been proven to reduce antibiotic utilization in respiratory diseases, Jeger postulated that PCT could do the same for diverticulitis [65,66].

Jeger et al. [64] demonstrated that PCT levels may help differentiate uncomplicated from complicated diverticulitis. Procalcitonin had been shown to have better specificity and sensitivity than CRP, especially in segregating out complicated from uncomplicated diverticulitis. However, their AUCs were not significantly different. Jeger showed that PCT alone was still inferior to CT scans because, in patients with complicated diverticulitis, 9% of cases had procalcitonin levels < 0.1 µg/L. Moreover, standalone PCT levels at admission had inadequate diagnostic capability, rendering a serial PCT level on day 2 necessary. This was attributed to the biological profile of PCT [64]. Jeger concluded that PCT measurements could not replace CT scans, nor should they be used as a standalone one-off parameter. However, PCT still seems to have a potential role in guiding antibiotic therapy in uncomplicated cases. Jeger further theorized that perhaps up to an 80% reduction in antibiotic therapy might be obtained with a PCT-guided treatment algorithm. This was extrapolated from studies in respiratory infections, where such reductions of antibiotic consumption were documented [64].

### 3.5. PCT-Guided Antibiotic Stewardship in Critically Ill Surgical Pts Reduces Antibiotics Use and Increases Survival

There is a fierce ongoing debate with regard to the ideal duration of antibiotics to treat intra-abdominal infections (IAIs) [67]. In general, IAIs can be divided into complicated and uncomplicated [68]. It is generally accepted that for cases of uncomplicated IAIs (e.g., cholecystitis), once surgery has been performed successfully, post-operative antibiotics are not needed. However, for complicated infections or cases where there is concern that the patient could develop septic shock, these patients should always be monitored closely using clinical parameters and inflammatory markers.

Some of the most well researched markers are CRP and PCT. Although testing for CRP is cheap, rapid and easily available, its levels are non-specific. Raised CRP levels can be associated with surgery, trauma, inflammation and infections of all kinds (viral, bacterial) [69].

On the other hand, PCT has been seen to be a more useful biomarker and has been shown to help in deciding the duration or escalation of antibiotics in septic patients [70]. PCT can also help in differentiating between SIRS and sepsis to some extent [71]. PCT has also been reported to be effective in guiding antibiotic treatment in critically ill patients [15]. Hochreiter et al. [10] conducted a trial to analyze the role of PCT in patients in the surgical intensive care unit (SICU), who were receiving antibiotics. In this study, 110 SICU patients were randomized to two groups. One group received a PCT-guided antibiotic protocol while the other group received a standard antibiotic regimen. Fifty-seven patients in the PCT group received antibiotics, whereby the regimen was regularly adjusted, as guided by daily PCT and clinical assessment. Conversely, there were 53 patients in the standard group. This group received a fixed standardized 8 day course of antibiotics. The trial concluded that those in the PCT group had received a significantly shorter course of antibiotics, as compared to the control group, without similar overall clinical outcomes. This demonstrated that PCT was a clinically useful tool for guiding antibiotic therapy, and can enable patients to avoid an unnecessarily prolonged therapy. This ultimately will help to reduce microbial resistance and costs overall.

A meta-analysis by Prkno et al. [72] included seven studies, with a combined total of 1075 patients with severe sepsis or shock. This paper studied outcomes between patients who received PCT-guided antibiotics versus those who received standard courses of antibiotics. Overall, both groups had similar hospital and 28-day mortality rates with relative risks of 0.91 and 1.02, respectively. Both groups also had similar length of stays in the ICU and hospital. This is despite the PCT-guided group having been given a significantly shorter course of antibiotics with a 1.27 hazard ratio.

Carr et al. [73] performed a review on four randomized trials studying septic patients in ICU settings [10,11,74,75]. In all four trials, a PCT-based algorithm was used to guide antibiotic therapy. The antibiotic duration was determined by serially trending elevated PCT levels until these levels decreased below a certain cut-off point. At that point, antibiotics were stopped. However, the PCT levels used to determine antibiotic cut-off were different across the four trials. In two trials, the cut-off level adopted was PCT < 1 ng/mL or <35% of the initial PCT level, within 72 h [10,11]. In the third trial, the cut-off level adopted was PCT < 0.25 μg/L or a decrease of >90% of the peak level. In the fourth trial, the cut-off was PCT < 0.5 μg/L or a decrease of >80% of the peak concentration [74,75].

Carr reported that despite the differences in cut-off values, all four trials drew similar conclusions. Most notably, all trials demonstrated that the PCT-guided groups experienced significantly shorter durations of antibiotic therapy, with a median reduction ranging from 1–3.5 days [10,11,74,75]. The Schroeder study also showed that a PCT-guided regimen had significant cost reductions of 18%. The Nobre trial further demonstrated that the PCT group had shorter lengths of stay in the ICU [11,74]. Despite the shorter duration of antibiotics, all trials reported similar clinical outcomes.

Carr, however, raised one major worry with a PCT-guided therapy algorithm. Carr was concerned that a septic patient might have had premature stoppage of antibiotics, which could lead to a recurrence of infection or other sepsis-related morbidity. This concern was addressed in two of the trials [74,75]. The Nobre trial [74] demonstrated that there were no recurrent infections in the PCT group once PCT levels had plummeted below 0.25 μg/L. The PRORATA trial showed that there were no deaths in the PCT group, after stoppage of antibiotics when PCT levels were less than 0.5 μg/L [75].

The key question remains regarding the cut-off value to be adopted. The Nobre trial showed that a cutoff level of 0.25 μg/L was adequate [74]. Another trial showed, however, that a higher cut-off level of 0.5 μg/L was also safe [75]. However, if the cut-off level was set too low at 0.1 ng/mL, then no significant benefit was observed.

Carr surmised that the appropriate cut-off level to stop antibiotics in septic ICU patients was a PCT level < 0.5 ng/mL. This threshold was successfully used in two trials [8,74]. The other two trials reported even higher levels of 1.0 ng/mL [10,12]. Nonetheless, the 0.5 ng/mL level was successfully used as an initial threshold not to start antibiotics upon enrolment in three trials [8,75,76].

Wirz et al. [77] performed a meta-analysis of 11 randomized trials, with a total of 4482 patients. Wirz demonstrated an overall lower mortality associated with PCT-guided therapy, similar to the results of another Dutch trial [78]. This effect was consistent across all septic patients, regardless of severity or type of sepsis. The PCT-guided therapy also had shorter antibiotic durations and earlier stoppage of antibiotics. However, PCT guidance did not have an effect on length of ICU or hospital stay.

### 3.6. Patients with Secondary Peritonitis Can Benefit from a PCT-Guided Regimen

Some studies have observed that PCT levels were correlated to prognostic outcomes in patients suffering from secondary peritonitis. These studies have proposed that perhaps PCT could be used as an indicator of successful treatment outcomes when managing these cases of intra-abdominal sepsis [16,79].

Huang et al. published a study that showed that a PCT-guided algorithm reduced antibiotic use [80]. In this study, patients who underwent emergency surgery for secondary peritonitis were prospectively studied. PCT concentrations were taken just before surgery and on post-operative days 1, 3, 5 and 7. Antibiotics were stopped when PCT levels measured <1.0 ng/mL or if PCT levels dropped by >80%, with clinical improvement of the patient. Huang subsequently performed propensity score matching against historical controls. Huang et al. demonstrated that the PCT group had significantly reduced duration of antibiotics, with a median duration of 3.4 days versus a median of 6.1 days for the control group. Overall clinical outcomes were comparable. Further analysis on reduction in hazard of antibiotic exposure demonstrated that the PCT group had an 87% reduction within 7 days and a 68% reduction after 7 days [80].

Maseda et al. [81] reported that, in 121 patients with secondary peritonitis, outcomes were similar between the PCT-guided and the standard therapy groups in terms of SICU or hospital length of stay, 28-day or in-hospital mortality. However, there was a 50% decrease in mean antibiotic exposure days in the PCT group.

Large trials have demonstrated that, even in severely septic patients in the ICU, the use of PCT guidance leads to shortened antibiotic durations [10,11,67,74,75,82,83]. Two systematic reviews have shown that PCT guidance led to a significant decrease in antibiotic duration in septic ICU patients, without an increase in adverse events [72,84]. These findings also extend to community-acquired secondary peritonitis [80].

In most of these PCT-based treatments, a classical cut-off of 0.5 ng/mL or higher [10,76,85] was used for surgical patients [72,83]. Some papers have also advocated that an 80% reduction from peak PCT levels could be utilized as a cut-off, especially if these patients started out with very high initial PCT levels [72,76,86].

### 3.7. PCT Is Useful in Identifying Patients at Risk of Complications after Major Abdominal Operation, Especially for Elective ERAS Patients, and Can Delay Discharge Even If Clinically Well

Spoto et al. [87] reported on 90 patients who underwent major abdominal surgery. PCT levels were taken pre-operatively and on each of the first three post-op days. Spoto et al. reported that those patients with PCT levels >1.0 ng/mL on post-op day 1 or 2 and >0.5 ng/mL on post-op day 3 were likely to have post-operative complications such as infections. Conversely, patients who had PCT levels < 0.5 ng/mL on post-op day 5 were fit and safe for an early discharge.

One possible explanation for this PCT trend was that, after a physiological insult, the plasma levels of PCT start to climb within 6 h and plateaus within 8 to 24 h. However, this level will decline dramatically when the insult has been removed. This rapid bio-kinetic action could be applied to spot patients who might have post-operative infections, as early as the first 24 h post surgery [88]. This is corroborated by another study that showed that PCT values ≥ 0.5 ng/mL after the first 24 hours after surgery are suggestive of either a possible post-surgical infection or inadequate source control at the index surgery [89].

Meisner et al. [90] also reported that, after major abdominal surgery, if PCT values were high on post-op days 1 or 2, this could suggest bacterial contamination. This could have occurred during the procedure itself or during anastomosis creation. Any contamination, regardless of cause, could predispose to surgical site infections and persistent sepsis. Therefore, it was possible that the immediate post-surgery PCT levels could indicate how dirty or contaminated a procedure had been, and hence the risk for infectious complications.

Of debate however is the PCT cut-off value. Typically, this is taken to be 0.5 ng/mL. This cut-off could, however, be overly sensitive in the setting of abdominal surgery. Mokart et al. [91] suggested that, for these cases, the cut-off for PCT should be set at 1.1 ng/mL, if measured on post-op day 1. Any reading above this on post-op day 1 would suggest a possibility for post-surgical infections. Spoto et al. [87] proposed a similar cut-off value of >1.0 ng/mL on post-op day 1. On the second post-op day, Oberhofer et al. [92] suggested using a cut-off value of 1.34 ng/mL.

On post-op day 5, Garcia-Granero et al. [93] suggested a cut-off of 0.31 ng/mL, citing that this level produced 100% sensitivity, 72% specificity, 100% negative predictive value and 17% positive predictive value. Garcia-Granero et al. further concluded that PCT was the most suitable biomarker, as it had an AUC of 0.86. Another study corroborated these findings, albeit suggesting that the post-op day 5 cut-off should be set at 0.5 ng/mL [90]. In the subgroup of 53 patients, this cut-off showed a high negative predictive value and allowed for early and safe discharge of patients [87].

Conversely, high levels or increasing trends of PCT during the first 72 hours after surgery could indicate that the patient was at risk for post-surgical infectious complications. For this group, a close and intense clinical, microbiological and early imaging evaluation should be recommended [87].

Being able to recognize patients with potential early post-operative issues is particularly helpful in the era of Enhanced Recovery After Surgery (ERAS). ERAS has been shown to be effective at reducing complication rates by approximately 40%, shortening length of stay (LOS) and reducing costs [94,95,96]. However, due to shortened LOS, the period of post-operative observation has likewise been cut short. This carries the inherent risk that some complications may manifest only after discharge, which would lead to a mandatory readmission [97].

Most post-operative surgical site infections are usually clinically apparent 4–6 days after the surgery. This exceeds the typical ERAS-driven length of stay. Therefore, it has become paramount to identify markers and parameters which could provide early detection of infectious complications [98]. This would provide a more objective and clinically sound method to decide if a particular patient should either continue to stay in hospital, or to at least have close interval monitoring soon after discharge.

Several studies have analyzed an array of biomarkers such as CRP, interleukin-1 (IL-1) and -6 (IL-6), or PCT. All these markers have demonstrated variable extents of clinical utility in predicting post-operative infections [99,100,101,102]. However, dedicated studies of the application of these markers in ERAS patients remain scarce and few [103,104].

Wierdak et al. [105] reported that, among patients with uncomplicated post-operative courses, levels of CRP and PCT would demonstrate a rapid rise on post-op day 1. Thereafter, these values would steadily decrease over the following days. This trend was not seen in patients who developed post-operative complications. Statistically significant differences in PCT levels between both groups were observed as early as post-op day 1. The analyses of ROC curves concluded that the greatest utility of this parameter was on post-op day 3. This is consistent with other studies that demonstrated PCT’s significance and utility in colorectal surgery, especially in recognizing anastomotic leaks and other surgical site infections [106,107,108].

Hayati et al. [109] showed that, in colorectal surgery patients, PCT had a negative predictive value of 100%. Hence, in a patient with a normal PCT value on post-op day 3, any suspicion of an anastomotic leak could be safely ruled out.

El Zaher [110] also reported that the PCT trend was the best predictor of major anastomotic leaks (AL). The PCT values were taken on post-op days 1, 3 and 5. On post-op day 5, when applying a PCT cut-off level of 4.93 mg/L, El Zaher demonstrated that PCT had a 97.3% negative predictive value with an AUC of 0.89. When PCT was combined with the 5-day trends of CRP and WBC, it gave a maximum discriminatory value for the detection of leaks. A meta-analysis by Cousin et al. also confirmed that PCT, measured on post-op day 5, was a useful biomarker for the early diagnosis of intra-abdominal infections, including leaks, after colorectal surgery [111]. Conversely, Giaccaglia et al. showed that low levels of PCT on post-op days 3 and 5 were associated with a low risk of anastomotic leaks [107].

### 3.8. PCT and Diagnosis of Infected Pancreatic Necrosis

The World Society of Emergency Surgery (WSES) guidelines recommended that procalcitonin was the most sensitive indicator when used to assess for infected pancreatic necrosis, and that low levels of PCT were strong negative predictors of infected pancreatic necrosis [112]. This is despite the traditional teaching that considers CRP as the preferred marker [113].

CRP has demonstrated that its sensitivity and specificity range from 38 to 61%, and 89 to 90%, respectively [113]. Unfortunately, CRP only reaches its maximal peak levels 48–72 h after admission.

In a study of 175 patients, both CRP and IL-6 produced AUCs of 0.803 [114]. Other laboratory parameters that were useful for predicting infected necrosis include blood urea nitrogen > 20 mg/dL, hematocrit > 44%, lactate dehydrogenase and PCT [113,115,116,117,118]. Within the first 96 h of symptom onset, if a PCT cut-off level of 3.8 ng/mL is adopted, PCT can detect pancreatic necrosis with a 93% sensitivity and 79% specificity [113,117].

### 3.9. PCT Levels Can Have Prognostic Value in the Initial Management of Patients, Even in the Emergency Department (ED)

Whilst data on PCT as a tool in the management of patients with IAI are sparse, there are even fewer data on how early PCT levels should be taken. In this regard, Covino et al. [119] demonstrated that, in the emergency department, PCT levels > 0.5 ng/mL were an independent risk factor for all-cause in-hospital mortality (HR 1.77 [1.27–2.48], and 1.80 [1.59–2.59], respectively). Fransvea et al. [120] also demonstrated that, in patients with acute cholecystitis, having a PCT value > 0.09 ng/mL upon ED admission had a sensitivity of 84.8% [68.1–94.9] and a specificity of 51.8% [43.2–60.3] for the occurrence of a major complication during the remainder of the stay.

## 4. Conclusions

There is increasing evidence to show that PCT is a very powerful biomarker in the management of intra-abdominal infections. A PCT-guided antibiotic stewardship program has been shown to effectively reduce the number of days of antibiotic treatment. However, much more research and many more trials need to be done, especially to discover the appropriate cut-off points and also to better understand the correct clinical situation in which to apply such algorithms.

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
