# Peer review of "Management of Intra-Abdominal Infections: The Role of Procalcitonin"

_antibiotics, 2023, doi:10.3390/antibiotics12091406_

Round 1
Reviewer 1 Report
Procalcitonin [PCT] is a valuable biomarker of sepsis. The diagnosis accuracy and specificity of PCT are higher than those of CRP. However, this manuscript dose not contain new perspectives.
Author Response
Thank you for your feedback. This paper was more of a systematic review, pooling together the sparse data on PCT and its role in managing intra-abdominal infections. A lot of surgeons may use PCT values in routine work; but many dont have a good grasp on the latest full body of evidence. Hence this paper is meant to pool data and give a quick glance and update.
Reviewer 2 Report
Blood tests to be used in the diagnosis of intra-abdominal infections and in guiding antibiotic therapy are important. Procalcitonin is promising in this respect. I think it is a valuable review that will contribute to the literature.
1. Diagnosis of intra-abdominal infections and management of antibiotic therapy are difficult. A compilation of studies in the literature in this area may be helpful to clinicians.
2. Handling intra-abdominal infections under separate headings provided a clearer understanding of the data.
3. The methodology part can be written in more detail. If there are publications that are not included, the reasons can be explained in more detail.
4. The discussion section is descriptive. References are sufficient.
5. In the reviewed articles, if there is a new molecule that can be an alternative to procalcitonin in the management of intra-abdominal infections, it can be mentioned in a few sentences.
Author Response
Thank you so much for your feedback.
the 2 major points that we would like to answer.
1) number of papers. there are quite a number of papers, however, alot of it do not mention abdominal infections. For those that do, it is part of a greater study. The dearth of papers makes it more important and vital that a review like our paper is published, so that others can have a sense on the latest data. but it would be hard to perform a thorough PRISMA based SR or even a MA.
2) we purposefully avoided other biomarkers because apart from CRP, the rest are poorly studied. Even more scarce data than PCT.
Reviewer 3 Report
See comments in the manuscript

Author Response
thank you. we have made adjustments in certain areas, taking into account other reviewers' and the managing editors' feedback.
Round 2
Reviewer 1 Report
PCT is a well-known biomarker of sepsis and can help in making early clinical decisions regarding management of patients.However, this manuscript dose not contain new perspectives.